# Analysis of individual differences in neurofeedback training illuminates successful self-regulation of the dopaminergic midbrain

Lydia Hellrung [1✉], Matthias Kirschner [2,3], James Sulzer[4], Ronald Sladky [2,5], Frank Scharnowski[2,5], Marcus Herdener[6] & Philippe N. Tobler[1]

The dopaminergic midbrain is associated with reinforcement learning, motivation and decision-making – functions often disturbed in neuropsychiatric disorders. Previous research has shown that dopaminergic midbrain activity can be endogenously modulated via neuro-feedback. However, the robustness of endogenous modulation, a requirement for clinical translation, is unclear. Here, we examine whether the activation of particular brain regions associates with successful regulation transfer when feedback is no longer available. More-over, to elucidate mechanisms underlying effective self-regulation, we study the relation of successful transfer with learning (temporal difference coding) outside the midbrain during neurofeedback training and with individual reward sensitivity in a monetary incentive delay (MID) task. Fifty-nine participants underwent neurofeedback training either in standard (Study 1 $N = 15$, Study 2 $N = 28$) or control feedback group (Study 1, $N = 16$). We find that successful self-regulation is associated with prefrontal reward sensitivity in the MID task ($N = 25$), with a decreasing relation between prefrontal activity and midbrain learning signals during neurofeedback training and with increased activity within cognitive control areas during transfer. The association between midbrain self-regulation and prefrontal temporal difference and reward sensitivity suggests that reinforcement learning contributes to suc-cessful self-regulation. Our findings provide insights in the control of midbrain activity and may facilitate individually tailoring neurofeedback training.

[1] Zurich Center for Neuroeconomics, Department of Economics, University of Zurich, Zurich, Switzerland. [2] Department of Psychiatric, Psychotherapy and Psychosomatics, Psychiatric University Hospital, University of Zurich, Zurich, Switzerland. [3] Division of Adult Psychiatry, Department of Psychiatry, Geneva University Hospitals, Geneva, Switzerland. [4] Department of Mechanical Engineering, University of Texas at Austin, Austin, TX, USA. [5] Department of Cognition, Emotion, and Methods in Psychology, Faculty of Psychology, University of Vienna, Vienna, Austria. [6] Center for Addictive Disorders, Psychiatric University Hospital, University of Zurich, Zurich, Switzerland. ✉email: lydia.hellrung@econ.uzh.ch

The dopaminergic midbrain, including the ventral tegmental area (VTA) and substantia nigra (SN), plays a crucial role in reward processing, reinforcement learning[1–4], motivation[5,6], and decision-making[7]. Dysfunctions of the reward system have far-reaching consequences and are associated with the development of psychiatric disorders, such as addiction[8] and schizophrenia[9,10]. Despite decades of extensive neuroscience and imaging studies which have contributed to an impressive body of knowledge of normal and abnormal reward system function, the neural mechanisms controlling midbrain activity are still not fully understood[11]. One key issue that has received increasing attention is whether humans are able to cognitively control brain activity within the reward system. It has already been shown that both healthy controls[12,13], and patients with cocaine addiction[14] can use visually displayed SN/VTA activity to learn to regulate it during real-time functional magnetic resonance imaging (rt-fMRI) neurofeedback training by means of mental imagery. However, the outcome of primary interest in neurofeedback training is a transfer beyond training itself, i.e., the ability to regulate activity also after training and without feedback.

Transfer of neurofeedback training is critical for clinical applications, including those involving disorders of the reward system[15], but the evidence is mixed. Specifically, MacInnes and colleagues[13] observed significant neural transfer effects following rt-fMRI training in the form of increased neural activity and connectivity of the VTA during transfer on the group level. However, other studies[12,14] using neurofeedback training of the SN/VTA found little group effects and substantial between-subject variation in self-regulation performance at transfer. Here, we go beyond the prior work by investigating how this variation arises, how individuals with more transfer success differ from those with less transfer success, and whether activity in brain regions other than the VTA characterizes individuals with successful transfer. We also ask whether individual differences in learning (defined as change in SN/VTA activity during training, when feedback is available) are related to individual non-midbrain transfer success (change in neural activity outside the SN/VTA in post-training session relative to baseline, both without feedback). Thus, the main contribution of our study is to characterize the neural mechanisms related to successful self-regulation of SN/VTA activity. Specifically, we combine data from two previous rt-fMRI neurofeedback studies[12,14] and pursue three aims.

(Aim 1) Our first goal was to characterize often neglected[16] individual differences in the degree of successful transfer of SN/VTA self-regulation and thereby differentiate regulators from non-regulators. We reasoned that the cognitive (or executive) control network is in a prime position for performing a demanding task such as imagery[17] and shaping subcortical brain regions. Indeed, animal and human studies have shown direct and indirect anatomical connections between prefrontal cortex and SN/VTA[18–21] and functional studies using electrophysiology[22], optogenetics[23], or fMRI[24] have corroborated the physiological relevance of prefrontal-SN/VTA interactions. Therefore, we test the hypothesis that successful transfer of SN/VTA regulation is associated with activation in brain regions that are part of the cognitive (executive) control network, especially prefrontal areas.

(Aim 2) Our second goal was to determine whether the framework of (operant) associative learning can be used to explain neurofeedback training. In applications of the associative learning framework to neurofeedback[17,25], the feedback provides a higher order reward and the chosen mental strategy is reinforced in proportion to the sign and magnitude of the feedback. At the beginning of the training, participants cannot predict which strategy will consistently lead to up- or downregulation of brain activity within the target region. Therefore, if they use an adequate strategy, participants receive more reward than predicted corresponding to a positive temporal difference between consecutive (actions in) states. As a consequence, they would be more likely to repeat the strategy, expect higher feedback next time and gradually learn how to keep the feedback signal within the dopaminergic midbrain high. Accordingly, in regulators the size of the temporal difference between feedback states should gradually decrease as the expected feedback increasingly converges with the actual feedback. In contrast, for non-regulators and participants in a control group receiving unrelated or unstable feedback, the temporal differences would remain large and variable because these participants cannot learn any association between mental strategies and feedback. These straightforward implications of current theorizing about the mechanisms underlying neurofeedback remained largely untested (for a simulation study on the temporal dynamics of feedback: Oblak and colleagues[26]; for the correlation of BOLD with signal increase (i.e. success) and decrease (i.e. failure) during regulation: Radua and colleagues[27]). Here, we investigate the temporal difference mechanism in regions that communicate with the SN/VTA by testing for non-midbrain associations with midbrain temporal difference signals. Note that the SN/VTA has been traditionally associated with the coding of reward prediction errors in both animal[2,28,29] and human research[30,31]. Furthermore, the causal sufficiency of dopaminergic prediction error signals for learning has been reinforced by optogenetics[32,33]. Together, we hypothesize that decreasing SN/VTA temporal difference signals during neurofeedback learning are associated with successful self-regulation as expressed in positive transfer success in non-midbrain regions. In other words, we test whether non-midbrain regions show a negative correlation with temporal difference signals as coded by the midbrain over the course of the experiment and relate this decrease to transfer success.

(Aim 3) Our third goal was to further distinguish regulators from non-regulators. Hence, we related the individual differences in the ability to self-regulate midbrain activity to individual markers of neural reward sensitivity. We asked whether successful self-regulation, as measured by transfer effects, taps into general properties of the reward system. Given that adaptive reward processing characterizes the SN/VTA[1,34] we used a variant of the monetary incentive delay (MID) task that captures differences in adaptive reward sensitivity between clinical and non-clinical populations[35]. Using this task, we tested the hypothesis that reward processing in regions that may control the dopaminergic midbrain is related to successful SN/VTA self-regulation.

In sum, to study individual differences in the capability to gain control of the SN/VTA we used rt-fMRI neurofeedback training in healthy participants receiving either positive-going feedback (standard group) or inverted (negative-going) feedback (control group). We quantified the individual degree of successful midbrain transfer by comparing the individual post-training versus pre-training self-regulation capabilities. Moreover, we related individual differences in reward sensitivity to separately measured midbrain transfer success.

## Results

**No difference in degree of regulation transfer (midbrain DRT) across groups**. We first evaluated the midbrain DRT measure (Fig. 1 and Supplementary Fig. 1) and compared it between the three datasets. There were no significant differences across all three groups (mean DRT standard group Study 1 = 0.01, mean DRT standard group Study 2 = −0.02, mean DRT inverted group Study 1 = −0.05. Both parametric ANOVA: $F(2, 56) = 0.13$, $p = 0.8$ and non-parametric Kruskal-Wallis: $H(2) = 0.39$, $p = 0.82$ tests concurred on differences not being significant). Moreover,

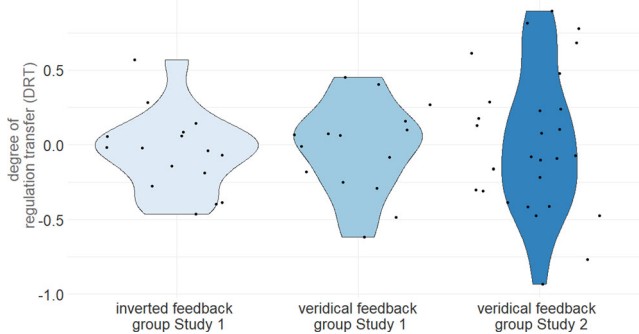

**Fig. 1 Midbrain DRT is distributed similarly across groups.** The midbrain DRT measure was distributed similarly in both groups receiving standard feedback in Studies 1 ($n = 14$ subjects) and 2 ($n = 28$ subjects) and the control group receiving inverted feedback in Study 1 ($n = 17$ subjects). Accordingly, we found no evidence supporting a main effect of feedback type on transfer. However, midbrain DRT varied substantially across individuals, which motivated the analyses of non-midbrain DRT, assessing correlations with individual midbrain DRT at the whole-brain level.

also the direct comparison between the two standard feedback groups was not significant (T(39) = −0.26, $p = 0.8$). Accordingly, we combined the two standard groups for subsequent analyses. Importantly, our participants showed considerable variation in DRT, which allowed us to investigate the individual differences in brain activity accompanying more or less successful transfer of SN/VTA self-regulation through neurofeedback. Thus, the groups showed similar mean levels and considerable individual differences in transfer success.

**Correlation of slopes between transfer success and self-regulation during training only for standard group (manipulation check).** As a manipulation check, we investigated the relation between training and transfer success. Specifically, we determined the slope of SN/VTA signal increase over training (i.e., the averaged difference of IMAGINE_REWARD – REST blocks in the second neurofeedback training run – first neurofeedback training run) and related it to midbrain DRT in Spearman's correlations for the standard and inverted feedback group. We found positive correlations between the slope of SN/VTA signal change during training period and midbrain DRT only for the standard feedback group, but not for the control group (Supplementary Fig. 2; standard group $\rho = 0.62$, $p < 0.001$; inverted group $\rho = −0.3$, $p = 0.25$, one-sided test for difference between correlations in independent samples z = 3.03, $p = .001$). Although the comparability between training and transfer is limited due to the feedback signal processing, particularly those individuals who were more successful at transfer also showed stronger upregulation during training and this relation was specific to the standard feedback group.

**Individual variation in transfer: midbrain DRT associated with cognitive control network in standard and amygdala activity in inverted feedback group (aim 1)**

*Standard feedback group.* We investigated whether individual levels of successful SN/VTA self-regulation (midbrain DRT) were associated with increased post- minus pre-training activity in other regions of the brain, i.e., non-midbrain DRT. This analysis revealed several areas consistently reported by neurofeedback studies (see Fig. 2 in the literature review of Sitaram et al.[17]), including dorsolateral prefrontal cortex (dlPFC), anterior cingulate cortex (ACC), lateral occipital cortex (LOC), and thalamus (Fig. 2a and Table 1). To formally test for a more general

association with the cognitive control network, we applied a cognitive control network template from a meta-analysis[36], which in addition revealed neural activity in precuneus and striatum (Fig. 2b for exemplary illustrations of dlPFC, ACC, temporal gyrus, and thalamus activity; Supplementary Table 1 for full overview). Thus, regions of the cognitive control network showed transfer to the extent that neurofeedback training of the dopaminergic midbrain was successful.

*Inverted feedback group.* For the inverted feedback group, the same analysis resulted in partly distinct activations. In contrast to the standard feedback group, left amygdala activity correlated significantly with midbrain DRT (Fig. 3 and Supplementary Table 2). Importantly, activity in cognitive control areas identified for the standard feedback group, such as dlPFC and ACC, was significantly weaker in inverted than standard feedback groups (Supplementary Table 3 for disjunction and direct statistical comparison). Together with the lack of correlation of midbrain DRT with SN/VTA signal change during training for the inverted feedback group, these findings suggest that cognitive control regions play a preferential role for successful transfer of SN/VTA self-regulation.

We also tested for common activity in the two feedback groups using conjunction analysis. Similar to the standard group, the inverted feedback group showed correlations between midbrain and non-midbrain DRT in the precuneus, middle temporal gyrus, insula, thalamus, and parahippocampal gyrus (Supplementary Table 4). These common areas appear to reflect non-specific regulation activity and may be associated with memory and introspection processes.

**Reinforcement learning: Reduced relation between dlPFC and midbrain temporal difference signals during neurofeedback training correlates with midbrain DRT (aim 2).** To investigate whether reinforcement learning mechanisms contribute to successful self-regulation transfer, we assessed the temporal differences in the SN/VTA feedback signal as proxy for the temporal difference signal during the neurofeedback training runs. We reasoned that temporal differences should decrease from early to late phases of neurofeedback training (at least in successful regulators, see next paragraph). Thus, we assumed that at any time during neurofeedback training, participants came up with their own predictions of the upcoming feedback signal and compared the predictions with actual feedback at the next time point. Similarly, in temporal difference learning models, errors in reward prediction are calculated at each moment in time[37]. Therefore, we operationalized these temporal differences by subtracting the immediately preceding SN/VTA activity (prediction) from the present SN/VTA activity (outcome: Supplementary Fig. 3). The basic contrast of temporal difference coding, i.e. without correlation to midbrain DRT, revealed striatal activity, a well established finding in the learning literature[38,39].

Next, we tested for a negative correlation of non-midbrain DRT with the difference in SN/VTA temporal difference signals between late and early training. In other words, for successful regulators, we expected to find a negative correlation between the decrease in midbrain temporal difference signal over the course of the neurofeedback training and activity in other brain regions. We found such a relation with gradually decreasing SN/VTA temporal difference signals in dlPFC (Fig. 4 and Supplementary Table 5). To interrogate the finding in more detail, we also analysed the two neurofeedback training runs separately. This analysis confirmed that only successful regulators showed a less pronounced relation between dlPFC activity and SN/VTA temporal difference error signals in late compared to early

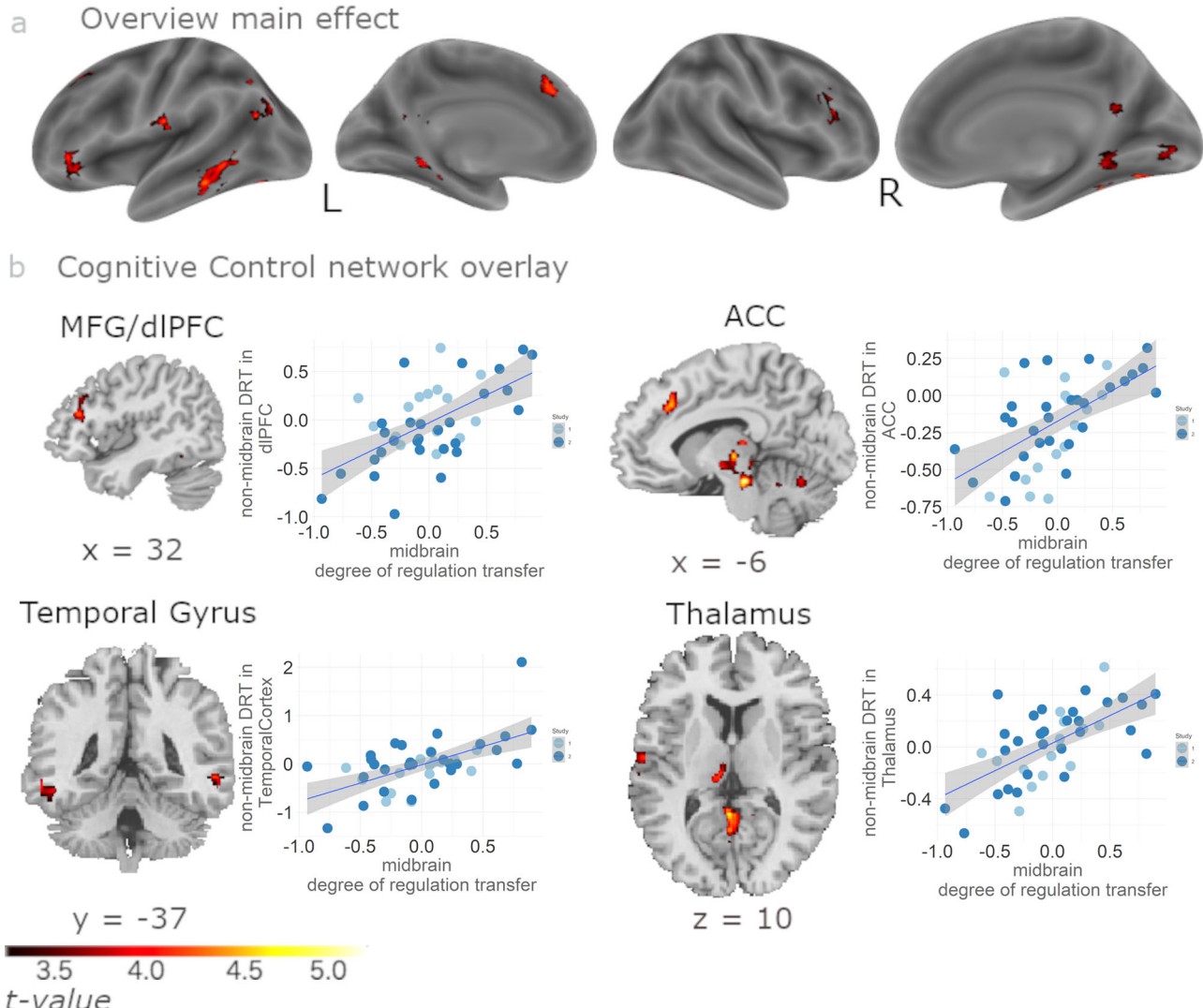

**Fig. 2 Midbrain and non-midbrain DRT correlate with cognitive control network in standard feedback group.** To investigate whole-brain neural activity correlating with successful SN/VTA self-regulation, we used midbrain DRT as measure of successful regulation of the SN/VTA and correlated it with non-midbrain DRT (DRT: IMAGINE_REWARD$_{transfer}$ - REST$_{transfer}$) – (IMAGINE_REWARD$_{baseline}$ – REST$_{baseline}$). **a** The analysis revealed correlations primarily within the cognitive control network (whole-brain overview FWE-corrected with $p < 0.05$ at cluster level, projected to lateral and medial sagittal sections). **b** Exemplary correlations within the cognitive control network in MFG/dlPFC, ACC, thalamus, and bilateral temporal gyrus, illustrating the association between midbrain DRT and non-midbrain DRT. The correlations are for illustration purposes only without further significance testing to avoid double dipping. The grey shaded area identifies the 95% confidence interval. See Table 1 for full result overview ($n = 42$ subjects).

training (see Supplementary Fig. 4 for run-wise analysis in dlPFC). Importantly, it should be noted that this decrease in the relation with learning-related error signals correlated with SN/VTA regulation success.

**Learning-related functional coupling of dlPFC with SN/VTA.** Following on from our finding of the decreasing relation between SN/VTA temporal difference error coding and dlPFC activity being indicative of individual success of regulating the dopaminergic midbrain, we performed a functional connectivity analysis to investigate whether the identified dlPFC region is related to the SN/VTA region our participants aimed to regulate. Thus, we used the dlPFC region showing decreasing relation with temporal difference coding during training particularly in successful regulators as a seed region and investigated whether it was coupled to the SN/VTA. Functional connectivity between the two regions increased with transfer success (Fig. 5; $t(40) = 3.79$, cluster extent = 16, MNI $x = -2$, $y = -16$, $z = -15$). In other words,

midbrain DRT and dlPFC to SN/VTA connectivity correlated positively. Note that this correlation of midbrain DRT with dlPFC-SN/VTA connectivity was task-related as it was enhanced during IMAGINE_REWARD relative to REST (which served as psychological regressor) and independent of SN/VTA activity.

**Individual differences in dlPFC reward sensitivity during MID task correlate with transfer success (aim 3).** In Study 2 we used the MID task to independently measure reward sensitivity and the capability to adapt to different reward contexts[35]. We asked whether individual measures of reward processing (measured with parametric and adaptive coding of reward related BOLD activity) are related to individual success in regulating the SN/VTA. Specifically, we tested for correlations between DRT and (i) MID reward sensitivity (sum of small and large reward parametric modulators) and (ii) MID adaptive reward coding (difference of small minus large reward parametric modulators). These two correlations both identified dlPFC (Fig. 6a). Moreover, a

**Table 1 Correlation of midbrain DRT with non-midbrain DRT in standard feedback group (see Fig. 2a).**

| Region label | # voxels | t-value | MNI coordinates | | |
|---|---|---|---|---|---|
| | | | x | y | z |
| Cingulate Gyrus, posterior division | 895 | 6.104 | −3 | −49 | 7 |
| Middle Frontal Gyrus (dorsolateral prefrontal cortex) | 295 | 4.609 | 45 | 31 | 19 |
| Left Thalamus | 1281 | 4.472 | −9 | −24 | 10 |
| Temporal Occipital Fusiform Cortex | 715 | 5.858 | 32 | −46 | −22 |
| Lingual Gyrus (Right Parahippocampal Gyrus) | 715 | 3.725 | 20 | −46 | −6 |
| Right Cerebral White Matter (Right Hippocampus) | 298 | 5.300 | 20 | −18 | −10 |
| Left Hippocampus | 408 | 3.703 | −26 | −33 | −12 |
| Left Cerebral White Matter (Left Middle Occipital Gyrus) | 693 | 5.056 | −36 | −73 | 31 |
| Left Cerebral White Matter (Left Superior Medial Gyrus, Anterior Cingulate Cortex) | 579 | 4.960 | −9 | 28 | 38 |
| Intracalcarine Cortex (Right Lingual Gyrus) | 856 | 4.048 | 6 | −82 | 1 |
| Occipital Fusiform Gyrus | 474 | 4.928 | 29 | −66 | −12 |
| Middle Temporal Gyrus (Left Inferior Temporal Gyrus) | 1028 | 4.913 | −62 | −37 | −16 |
| Middle Temporal Gyrus (Left Inferior Temporal Gyrus) | 1028 | 4.315 | −59 | −57 | −3 |
| Occipital Fusiform Gyrus | 658 | 4.866 | −42 | −70 | −19 |
| Temporal Fusiform Cortex, posterior division | 658 | 4.638 | −39 | −43 | −28 |
| Temporal Occipital Fusiform Cortex | 658 | 4.401 | −23 | −66 | −19 |
| Superior Frontal Gyrus | 310 | 4.736 | 6 | 2 | 80 |
| Central Opercular Cortex (Left Superior Temporal Gyrus) | 309 | 4.636 | −59 | −16 | 16 |
| Left Cerebral White Matter (Left inferior Frontal Gyrus) | 401 | 4.589 | −39 | 35 | −9 |
| Location not in atlas | 408 | 4.578 | −14 | −49 | −22 |
| Location not in atlas (Right Paracentral Lobule) | 341 | 5.003 | 2 | −39 | 82 |
| Location not in atlas (Left Cerebellum IV) | 856 | 4.947 | −6 | −66 | −15 |

The table shows all local maxima separated by more than 20 mm; for all clusters, $p < 0.05$ FWE-corrected on cluster level; df = 40. Regions were labelled using the Harvard-Oxford atlas and/or the Anatomy Toolbox in parentheses; x,y,z = Montreal Neurological Institute (MNI) coordinates in the left-right, anterior-posterior, and inferior-superior dimensions, respectively.

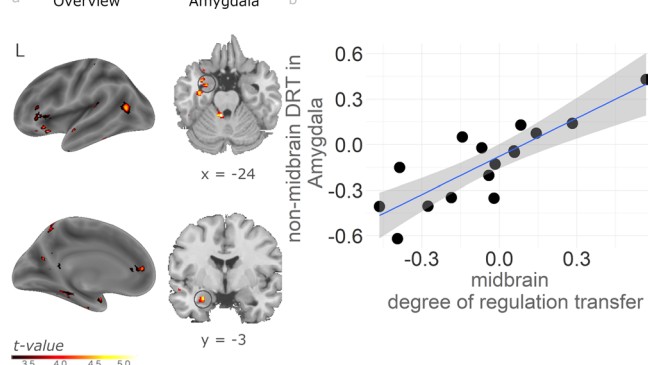

**Fig. 3 Midbrain and non-midbrain DRT correlate in amygdala in inverted feedback group. a** Receiving inverted feedback resulted in a correlation between midbrain and non-midbrain DRT ($\text{IMAGINE\_REWARD}_{transfer}$ − $\text{REST}_{transfer}$) − ($\text{IMAGINE\_REWARD}_{baseline}$ − $\text{REST}_{baseline}$) in the amygdala ($p < 0.001$ uncorrected). This region was not observed in the standard feedback group. **b** Scatter plot depicting the positive association between amygdala and midbrain DRT. The plot is for illustration purposes only without further significance testing to avoid double dipping. The grey shaded area identifies the 95% confidence interval. See Supplementary Table 2 for full result overview ($n = 17$ subjects).

conjunction of these two correlations with the correlation between midbrain DRT and non-midbrain DRT revealed common neural activity in the dlPFC (center at MNI $x = 40$, $y = 10$, $z = 38$; Fig. 6 and Supplementary Table 6). Thus, the more successful individuals were at self-regulating SN/VTA as a result of neurofeedback training, the more sensitive they were to reward and the more strongly they adapted to different reward contexts in the MID task.

## Discussion

In the present work, we used data acquired from two previous rt-fMRI neurofeedback studies to characterize individual differences and processes underlying successful transfer of self-regulation of the dopaminergic midbrain after neurofeedback training. This perspective on transfer success provided insights on what distinguished individuals who were more successful at SN/VTA regulation from those who were less successful: First, the midbrain degree of regulation transfer varies across individuals and this variance is related to different patterns of neural activity during neurofeedback training and transfer. Second, in particular, we found a significant relation between transfer success and increases in post- minus pre-training activity in the cognitive control network. Third, we found four correlations with stronger degree of midbrain regulation transfer: (i) decreasing relation between dlPFC activity and SN/VTA temporal difference error signals during neurofeedback training, (ii) stronger connectivity of dlPFC with the SN/VTA for reward imagination compared to rest during transfer, (iii) stronger reward sensitivity in dlPFC and (iv) stronger adaptive reward coding in dlPFC in the independent MID task. Fourth, control analyses revealed that alternative forms of learning (sensitization or desensitization) could not explain our findings.

Together, our study suggests that neurofeedback control of the dopaminergic midbrain relies on the cognitive control network and that the predictability of the upcoming feedback indexed by reinforcement learning signal contributes to successful neurofeedback training. Sustained self-regulation skills and the generalization of learning after neurofeedback training are key elements for practical applications and remain one of the major challenges in rt-fMRI neurofeedback research[40]. Pioneering neurofeedback studies of the reward system found little transfer on average[12,41,42]. Only one study[13] reported significant post-minus pre-training activity in the VTA, and increased mesolimbic network connectivity. This study used only the VTA as target

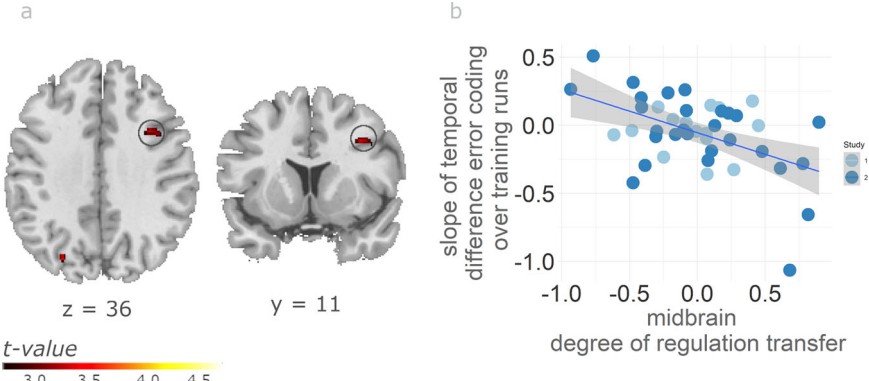

**Fig. 4 Reduced relation between dlPFC and midbrain temporal difference signals during neurofeedback training correlates with midbrain DRT. a** The relation between dlPFC activity and the SN/VTA signal encoding the temporal difference between the current and immediately preceding feedback activity decreased with ongoing feedback training (i.e., the slope over the first and second neurofeedback training run) more strongly in individuals with higher midbain DRT (p < 0.001 uncorrected). It is important to note that this figure depicts the negative correlation of temporal difference coding with DRT on the group level. The finding is consistent with reinforcement learning theories, according to which temporal differences decrease as learning progresses. By extension, a reinforcement learning framework can explain successful neurofeedback training. **b** Scatter plot illustrating that the slope of temporal difference signals in dlPFC over the NF training runs is statistically related to the individual degree of transfer success. The plot is for illustration purposes only without further significance testing to avoid double dipping. The grey shaded area identifies the 95% confidence interval (n = 42 subjects). See Supplementary Table 5 for full result overview.

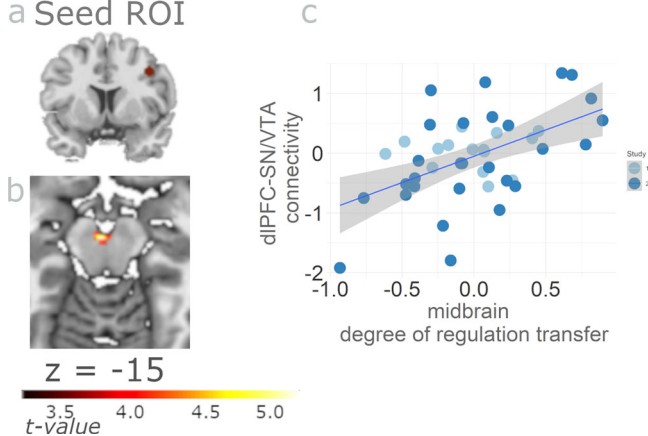

**Fig. 5 DlPFC and SN/VTA functional connectivity correlates with transfer success. a** Functional connectivity analysis based on the seed region in the dlPFC that coded the temporal difference error (MNI coordinate 40, 10, 38, and 5 cm sphere). Connectivity of that region with the SN/VTA correlated positively with success of neurofeedback training (p < 0.001 uncorrected). **b** Midbrain DRT increased with increasing connectivity between dlPFC and SN/VTA during IMAGINE_REWARD vs. REST in neurofeedback training runs. Thus, dlPFC appears to relate to SN/VTA in proportion to the degree to which neurofeedback training is successful. **c** Scatterplot illustrating the correlation between dlPFC -SN/VTA connectivity and midbrain DRT from (**b**). We perform no further significance testing to avoid double dipping. The grey shaded area identifies the 95% confidence interval (n = 42 subjects).

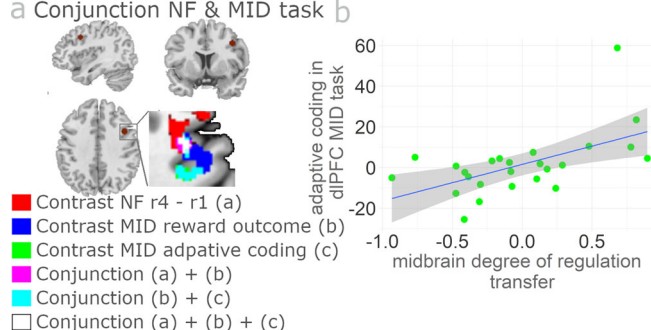

**Fig. 6 Reward-sensitivity in dlPFC correlates with successful SN/VTA self-regulation. a** Degree of successful SN/VTA transfer (midbrain DRT) in the neurofeedback task correlated with prefrontal reward sensitivity and adaptive coding in the MID task. A conjunction analysis around the peak coordinate in dlPFC showing midbrain DRT-related decreases in temporal difference coding during neurofeedback training (MNI x = 40, y = 10, z = 38, left) revealed common neural activity for non-midbrain DRT and general reward sensitivity (small + large reward magnitude parametric modulators in MID, all contrasts with p < 0.001 uncorrected). Moreover, individuals with more successful self-regulation of the SN/VTA showed stronger adaptive reward coding (which reflects higher sensitivity to small relative to large rewards) in the same region that also showed midbrain DRT-related decreases in temporal difference coding during neurofeedback training (right). **b** The scatter plot depicts the correlation between adaptive reward coding activity in dlPFC and midbrain DRT. The plot is for illustration purposes only without further significance testing to avoid double dipping. The grey shaded area identifies the 95% confidence interval (n = 25 subjects). See Supplementary Table 6 for full result overview.

region for neurofeedback and included visual and false-feedback rather than inverted control groups. The instructions given to the participants in this study differed from those of the other studies and our study mainly in the recommended strategies (see detailed comparison in Supplementary Note 1). These differences may partly explain why the latter study found a group effect but the others did not. Nonetheless, our findings yield the same direction on individual transfer success levels as the finding from MacInnes and colleagues (2016): successful regulators learn to increase dopaminergic signaling. We go beyond this work in several ways.

First, the previous studies focused exclusively on self-regulation of one a priori target region, such as SN/VTA, instead of investigating transfer effects within the whole brain. Second, transfer effects were examined at the group-level, which did not reflect the individual learning success. Therefore, in the present study we overcome these limitations by taking advantage of an individual measure of transfer success (midbrain DRT) and analysing the whole brain (non-midbrain DRT). Transfer success was distributed in our study, in line with studies using other

neurofeedback modalities such as EEG[16,43–46]. Our findings were specific to the SN/VTA as indicated by the control analysis with the parahippocampus and we avoided, or accounted for, potential confounding variables, such as practice and familiarity, personality measures and strategies. Thus, midbrain DRT as measured directly after neurofeedback training, meaningfully captures the individual capability of SN/VTA self-regulation.

One insight of the present study is that transfer success associates with non-midbrain neural activity in cognitive control network areas[36,47], such as dlPFC and ACC. The lack of cognitive control engagement within the control group and the correlation of midbrain DRT with the slope of SN/VTA increase during training in the standard feedback group suggests a preferential relation between successful transfer and learning. The cognitive control network includes regions that have been associated with feedback-related information processing during training[48,49]. Together, these findings suggest that the same regions contribute to acquisition and transfer of neurofeedback and that sustained post-training self-regulation generalizes across a functional network of different brain regions. Intriguingly, similar networks have been reported in skill learning and future studies may therefore investigate commonalities between neurofeedback and particularly cognitive skill learning, taking into account the specific temporal dynamics of both functions[25,50].

The finding that individuals with more successful regulation of the dopaminergic midbrain show stronger activation of cognitive control areas during transfer speaks to our understanding of how individual differences in cognitive control affect emotion regulation[51–54]. For example the working memory component of cognitive control has been shown to predict negative affect reduction through reappraisal and suppression[55]. Interestingly, dopamine action (particularly at D1 receptors) in dlPFC sustains working memory performance[56]. Thus, it is conceivable that frontolimbic loops contribute to successful transfer. In any case, this notion converges with our finding of dlPFC-SN/VTA coupling being related to transfer success.

Future research might explore whether the positive non-midbrain training effects in the cognitive-control network also have implications for transdiagnostic clinical applications. First, combining rt-fMRI neurofeedback training with different forms of psychotherapy such as cognitive behavioral therapy[57], dialectical behavioral therapy[58], or psychodynamic therapy[59–61] could improve emotion regulation deficits prevalent in several psychiatric disorders including substance use disorders, depression, anxiety and personality disorders. It has already been shown that in patients suffering from depression, neurofeedback training can be a successful tool to re-stabilize modulation of the amygdala and increase its responsivity to reward[62]. It remains a question for future patient studies if such training also enables re-stabilization of cognitive control. With particular attention to substance use disorders, maladaptive changes in neuroplasticity within the cognitive control network are closely associated with loss of control and compulsive drug-seeking[63–65]. In these patients, neurofeedback training might be able to directly target the biological correlates and reinstate function of the cognitive-control network and thereby of the SN/VTA.

While our first aim investigated a sustained form of dopaminergic responses during transfer, our second aim operationalized the first derivate of the sustained modulation in dopaminergic midbrain as temporal difference signal. This allowed us to investigate a more phasic form of responses outside the dopaminergic midbrain during training. We found a reduction in the relation between the decreasing SN/VTA temporal difference signal and dlPFC activity over the course of the neurofeedback training for successful regulators only, while this relation remained high for non-regulators. This finding suggests that a temporal difference-like signal might be tracked by the dlPFC, which is compatible with the notion that temporal difference error-driven reinforcement learning was more pronounced in regulators than non-regulators and provides empirical evidence for previous theoretical proposals on the principles of neurofeedback learning independent of feedback modality[25]. Thus, reinforcement learning provides a framework for understanding how neurofeedback works (by reducing temporal difference signals regarding future feedback). Future research may want to investigate whether the rich theoretical and empirical tradition of reinforcement learning[66] can be harnessed to facilitate neurofeedback training.

We found that successful SN/VTA self-regulation is associated with increased functional coupling between dlPFC regions related to temporal difference signals and the dopaminergic midbrain. This coupling fits well with anatomical connections between dlPFC and the dopaminergic midbrain[19,21] as well as effective connectivity studies on motivation[24] and animal studies on prefrontal regulation of midbrain activity[20,67]. While human work primarily focused on coupling between the prefrontal cortex and the striatum[68–70], animal work has documented both direct (glutamatergic) and indirect (through inhibitory interneurons) projections from prefrontal cortex to dopaminergic neurons[19,71,72]. These projections modulate event-evoked VTA activation consistent with phasic fMRI signals documented in humans[24], animals[22,73,74], and viral tracing/optogenetic stimulation studies[75]. Our data are in line with these findings and suggest comparable underlying mechanisms of dopamine release by volitional midbrain self-regulation.

At the functional level, a recent study on creative problem solving in humans associates dlPFC with experiencing a moment of insight[76]. According to this effective connectivity study, dlPFC could upregulate the VTA/SN via striatal connections during such a moment. On the other hand, in trials where no solution was found for a given problem, also no significant connectivity was observed. Our study reinforces the notion that dlPFC-SN/VTA connectivity plays an important role in self-guided motivation and in internal reward processing. Our finding points to the possibility that cognitive and affective mechanisms associated with different experiences also involve different neural pathways. Future studies may want to investigate to what degree individual differences in the functional architecture of brain networks[77] influence these internal reward mechanisms and to which degree different strategies can influence neurofeedback training success.

Our independent reward task revealed that individual differences in prefrontal reward sensitivity and efficient adaptive reward coding were associated with successful SN/VTA self-regulation. Adaptive coding of rewards captures the notion that neural activity (output) should match the most likely inputs to maximize efficiency and representational precision[78]. Accordingly, we previously showed that reward regions encode a small range of rewards more sensitively than the large range of rewards[79,80]. Interestingly, in the present study, participants who were more sensitive to small rewards were also more successful in self-regulation of the dopaminergic midbrain. When participants in a typical neurofeedback training paradigm succeed at increasing the activity of the self-regulated area, the ensuing change in visual stimulation (positive neurofeedback) may constitute a small reward. By extension, adaptive reward coding may therefore provide a useful handle on identifying regulators. Moreover, future neurofeedback experiments should consider scaling the feedback signal to avoid sensitivity limitations, particularly in individuals with reduced adaptive coding.

A potential limitation of our study is that we used a combined mask for SN and VTA even though differences in functionality and anatomy have been reported for the two regions (reviewed e.g. by Trutti et al.[81]), with the SN more related to motor

functions and the VTA to reward functions. However, it should be kept in mind that when viewed through the lens of recording and imaging rather than lesion techniques the differences between regions are more gradual than categorical[82]. Still, future studies may want to use more specific feedback from one or the other region to more specifically target potential differences in functions. Further limitations are that only inverted feedback is available here as control group and that this group has a smaller sample size. With regard to the interpretation of the results from the inverse feedback control, it is important to note that we controlled for explicitly stated strategies. But it is of course conceivable that some strategies are harder to become aware of and express in words than others and that this issue particularly applied to the inverted feedback group. One could speculate that the individual differences in the relation between amygdala and midbrain DRT in the control group reflect differences in frustration. Although a different brain network appears to be related to midbrain DRT in this group (Supplementary Fig. 5), this interpretation requires further investigation because the control group did not perform a tailored behavioral assessment of frustration after the experiment. As stated by Sorger and colleagues[83], including measures of frustration is considered critical in the design of neurofeedback studies nowadays. Additional control groups receiving no or noninformative feedback could help assess the effects of neurofeedback training in situations where participants cannot learn anything, although these control groups suffer from other issues, such as reduced contingency between regulation efforts and signal change, which may lead to disengagement[83]. Still, our data show a significant correlation between degree of regulation transfer and training runs only for the standard feedback group and not for the control group. Moreover, other neurofeedback studies have shown that volitional self-regulation of brain activity can only be learned when real feedback is presented[84] and that other control groups failed to acquire VTA self-regulation[13]. Nonetheless, future studies investigating such control groups will be necessary to replicate these findings. In general, we recommend for future studies to take current best practice guidelines[83,85] into account when designing neurofeedback experiments to overcome such limitations and maximize replicability of findings.

In conclusion, in a series of analyses integrating multiple fMRI measures and computational learning parameters we were able to parse out individual factors contributing to successful transfer of midbrain self-regulation after neurofeedback training. One such factor was activity in the cognitive control network, particularly dlPFC. Future studies could employ cognitive control activity during neurofeedback training to boost success rates and clinical outcomes. Furthermore, our findings suggest that associative learning contributes to real-time fMRI neurofeedback effects. Finally, we show that higher individual reward sensitivity in the dlPFC increases the chance of neurofeedback training success. Patients with reduced neural reward sensitivity may therefore benefit from careful scaling of the neurofeedback information to equalize the subjective value of the reward across participants.

## Methods

**Participants**. Fifty-nine right-handed participants (45 males, average age $28.25 \pm 5.25$ years) underwent SN/VTA neurofeedback training. We analysed data from two independent projects, which used highly similar rt-fMRI paradigms, rt-fMRI software and scanner hardware. The first dataset[12] comprised male participants, randomly assigned to one of two groups. The experimental group received three runs of standard neurofeedback ($N = 15$), the control group received inverted neurofeedback ($N = 16$) as training signal. The second dataset[14] comprised the healthy control participants ($N = 28$, 14 males) of a project investigating also cocaine users (these data are not presented here). This group received two runs of standard neurofeedback. A subset of the participants in the second dataset ($N = 25$) also performed a variant of the monetary incentive delay (MID) task[35]. In both studies, participants were recruited from the same age range of 24 to 35 years.

Study 1 recruited healthy, non-smoking participants from a departmental database of university students. In Study 2 healthy participants were recruited via online advertisement and matched on regard to sex, age, and nicotine consumption to an inpatient group of patients with cocaine addiction from the Psychiatric University Hospital. Exclusion criteria were clinically relevant somatic diseases, head injury or neurological disorders, family history of schizophrenia or bipolar disorder, and use of prescription drugs affecting the central nervous system. Additional exclusion criteria for both study groups were MRI ineligibility due to non-removable ferromagnetic objects in the body, claustrophobia, or pregnancy. All participants provided written informed-consent and received compensation for their participation. The Zurich cantonal ethics committee approved these studies in accordance with the Human Subjects Guidelines of the Declaration of Helsinki.

**Experimental setup and neuroimaging**. All participants underwent neuroimaging in a Philips Achieva 3 T magnetic resonance (MR) scanner using an eight channel SENSE head coil (Philips, Best, The Netherlands) either at the Laboratory for Social and Neural Systems Research Zurich (SNS Lab, Study 1) or the MR Center of the Psychiatric Hospital of the University of Zurich (Study 2). First, we acquired anatomical images (Study1: gradient echo T1-weighted sequence in 301 sagittal plane slices of $250 \times 250$ mm$^2$ resulting in 1.1 mm$^3$ voxels; Study2: spin-echo T2-weighted sequence with 70 sagittal plane slices of $230 \times 184$ mm$^2$ resulting in $0.57 \times 0.72 \times 2$ mm$^3$ voxel size) prior to neurofeedback training and loaded them into BrainVoyager QX v2.3 (Brain Innovation, Maastricht, The Netherlands) to identify SN/VTA as target region (see section on SN/VTA region-of-interest for details). For the functional scans, we used 27 ascending transversal slices in a gradient echo T2*-weighted whole brain echo-planar image sequence in both studies. The in-plane resolution was $2 \times 2$ mm$^2$, 3 mm slice thickness and 1.1 mm gap width over a field of view of $220 \times 220$ mm2, a TR/TE of 2000/35 ms and a flip angle of 82°. Slices were aligned with the anterior–posterior commissure and then tilted by 15°. Functional images were converted from Philips par/rec data format to ANALYZE and exported in real-time to the external analysis computer via the DRIN software library provided by Philips. This external computer ran Turbo BrainVoyager v3.0 (TBV – Brain Innovation, Maastricht, The Netherlands) to extract the BOLD signal from the images and calculate the neural activation for the feedback signal. The visual feedback signal was presented using custom-made software with Visual Studio 2008 (Microsoft, Redmond, WA, USA) through either a mirror mounted at the rear end of the scanner bore (Study 1) or through MR compatible goggles (Study 2).

**Neurofeedback paradigm**. The participants were instructed that their goal was to control a reward-related region in their brains by imagining rewarding stimuli, actions, or events. We have previously shown that reward imagination activates SN/VTA with conventional fMRI[86]. Prior to scanning, we provided examples of such rewards, including palatable food items, motivating achievements, positive experiences with friends and family, favourite leisure activity or romantic imagery. We encouraged participants to use these different rewards as potential strategies for upregulating reward-related activity during the cue 'Happy Time!', here referred to as IMAGINE_REWARD condition. In contrast, during the cue 'Rest' (here referred to as REST condition), participants were asked to perform neutral imagery, such as mental calculation to reduce reward-related activity. In both conditions, the real-time SN/VTA BOLD signal was continuously fed back to the participant in the form of a smiley that translated vertically in proportion to the signal (Fig. 7). Prior to training, participants were familiarized with the 5 s delay of the hemodynamic response affecting the display of the feedback and were asked not to move or change their breathing during the neurofeedback training. The control group received identical instructions and was debriefed after the session about the inversion of the feedback signal.

Each neurofeedback session comprised: a pre-training imagery baseline run without any feedback, three (Study 1) or two (Study 2) training runs during which neurofeedback was presented (as Study 2 also investigated patients, training was limited to two runs), and a transfer run (i.e., without feedback). Each of these runs comprised nine blocks of IMAGINE_REWARD and REST conditions, each lasting 20 s. To determine the current level of the feedback signal in the neurofeedback sessions, we used the average of the last five volumes of the previous REST condition as reference value (minimizing drift and motion effects) and employed a moving average of the previous three volumes to reduce noise. In the standard feedback group, the smiley moved up with increasing percent signal change in the SN/VTA BOLD signal and changed colour from red to yellow (Fig. 7a). In the inverted feedback group, the smiley moved up and turned yellow with decreasing SN/VTA BOLD signal.

**SN/VTA region-of-interest (ROI)**. In both studies, the target region for neurofeedback, i.e. the substantia nigra (SN) and ventral tegmental area (VTA), was structurally identified using individual anatomical scans. Since the individual mask definition slightly differed between Study 1 and 2 (T1-weighted scans in Study 1 and T2-weighted scans in Study 2), we used an independent mask for our post-hoc analysis. By this, we can control for individual differences between experimenter ROI selection strategies, avoid interpolation confounds due to warping by normalization and use a reliable seed region for functional connectivity analysis.

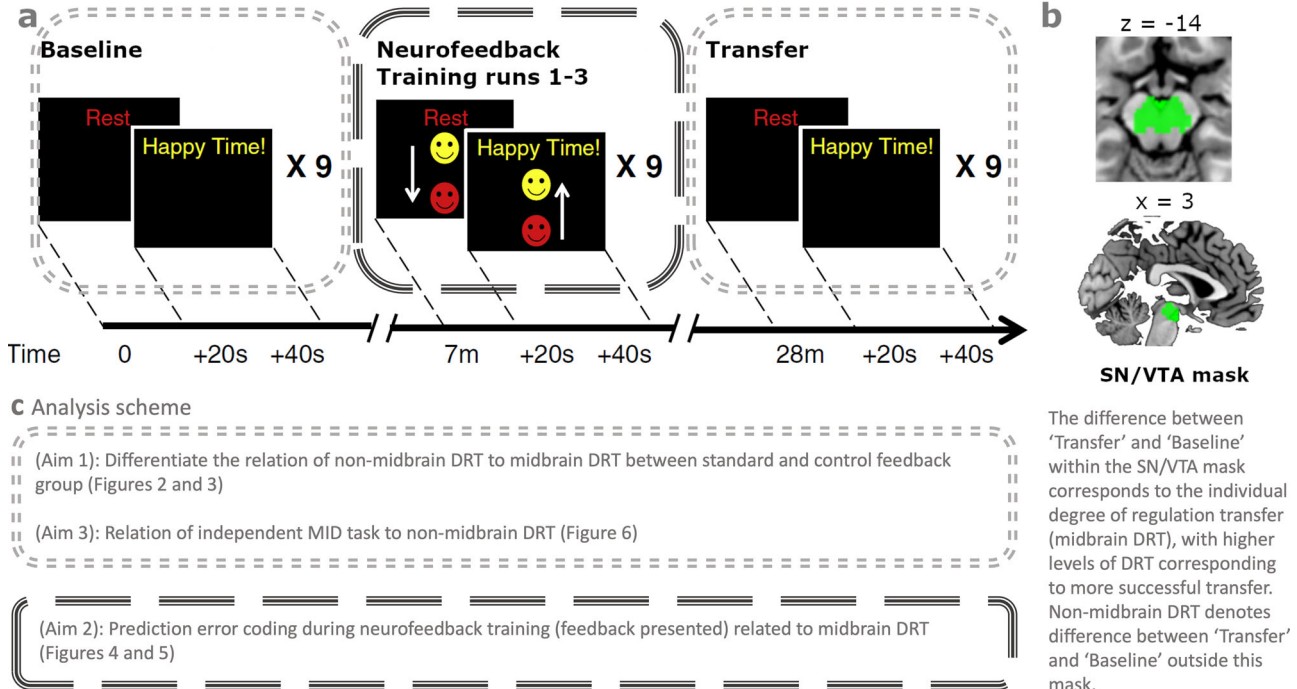

**Fig. 7 An overview of the neurofeedback paradigm. a** All runs consisted of alternating blocks of REST and IMAGINE_REWARD conditions, with each block lasting 20 s. The regulation conditions (REST, IMAGINE_REWARD) were indicated by words ('Rest' or 'Happy Time!') and the feedback presented as moving smiley face during neurofeedback training runs. The baseline and transfer runs comprised no feedback. The SN/VTA signal difference from these runs served to quantify the degree of regulation transfer (DRT) as (SN/VTA_BOLD_{IMAGINE_REWARD,Transfer} − SN/VTA_BOLD_{REST,Transfer}) − (SN/VTA_BOLD_{IMAGINE_REWARD,Baseline} − SN/VTA_BOLD_{REST,Baseline}). **b** Post-processed SN/VTA signal was extracted from the probabilistic atlas mask[98]. **c** The analysis scheme highlights partial data of the paradigm used to investigate the three aims of this study.

Specifically, we used the probabilistic mask of the SN and VTA as defined by[87], which is based on a large sample (148 datasets) and available on https://www.adcocklab.org/neuroimaging-tools (download August 2018). Figure 7b illustrates the mask within the brain. From this mask, we extracted and averaged SN/VTA activity for each participant using custom-made scripts in Matlab R2016b.

**Degree of regulation transfer (DRT).** We assessed the effects of individual differences in performance to characterise participants on a continuous regulation scale. The measure of successful self-regulation was defined as individual degree of regulation transfer (DRT). We calculated DRT both for the midbrain and non-midbrain regions, i.e. as the condition-specific signal difference between post-training (Transfer) and pre-training (Baseline) runs:

$$DRT = (BOLD_{\{IMAGINE\_REWARD,Transfer\}} - BOLD_{\{REST,Transfer\}})$$
$$- (BOLD_{\{IMAGINE\_REWARD,Baseline\}} - BOLD_{\{REST,Baseline\}}) \quad (1)$$

Thus, a positive DRT corresponds to a relative increase in post-training (midbrain or non-midbrain) BOLD activity compared to pre-training BOLD activity for the contrast IMAGINE_REWARD minus REST. It is essential to note that during these two runs (pre-training baseline, post-training transfer) no neurofeedback was presented. The absence of feedback ensures comparability between participants in the different groups, as the perception and processing of the feedback signal during the training runs might be different and influence the SN/VTA signal itself.

*Midbrain DRT distributions.* To investigate potential group differences in midbrain DRT, we transferred the extracted SN/VTA data to R (R-project R3.4.1). Using an ANOVA and a non-parametric Kruskal–Wallis test, we tested for differences between the three groups (i.e. the two groups receiving standard feedback in Studies 1 and 2 and the control group receiving inverted feedback in Study 1).

*Whole-brain correlations with midbrain DRT in fMRI analysis.* To investigate DRT outside the midbrain, we tested for individual differences in successful transfer at the whole brain level (non-midbrain DRT). Thus, non-midbrain DRT identified regions that were positively associated with midbrain DRT and thereby potentially contributed to regulation of the SN/VTA. This approach goes beyond previous research which contrasted average effects in healthy participants or in healthy controls against patient groups[12–14].

*Control analyses and measures.* To validate our approach of investigating individual differences in midbrain DRT and relating them to non-midbrain DRT, we performed several control analyses and took control measures. First, we performed a control analysis which assessed the spatial specificity of our effects. To do so, we correlated whole brain activity with a region other than the SN/VTA. Specifically, we used the neighbouring parahippocampus (Supplementary Note 2). In keeping with specificity, this control analysis revealed little commonality (limited to the cerebellum and temporal gyrus) with the SN/VTA analysis (Supplementary Fig. 6 and Supplementary Table 7). Note that if unspecific individual differences would explain our main findings then one would expect similar results in the control analysis, contrary to what we find. Second, we ascertained that midbrain DRT correlated with midbrain neurofeedback training. To achieve positive SN/VTA transfer effects, participants had to apply what they had learned during neurofeedback training runs. We therefore tested whether midbrain DRT was related to SN/VTA activity increase during the training runs by calculating the correlation between them. Specifically, we determined the slope of SN/VTA signal increase over training (i.e., the averaged difference of IMAGINE_REWARD – REST blocks in the second neurofeedback training run – first neurofeedback training run) and related it to midbrain DRT in Spearman's correlations for the standard and inverted feedback group. Third, we included only participants without prior experience with neurofeedback to control for practice and familiarity differences. Moreover, we performed a control analysis to check whether individuals with successful subsequent transfer differ already at the baseline session by correlating the IMAGINE_REWARD-REST contrast at baseline only with midbrain DRT. Using identical family-wise error cluster-level correction as for correlation with transfer-baseline revealed no findings within the cognitive control network at baseline only. Moreover, inclusively masking the baseline findings with transfer-baseline findings revealed no overlap of the two analyses. Thus, individual differences in midbrain DRT reflect primarily what participants learned during the neurofeedback training sessions rather than pre-existing individual differences. Fourth, we controlled for personality measures and strategies employed by participants. We note that we used (mean-centered) midbrain DRT as continuous scale with the aim of investigating the mechanisms underlying successful self-regulation performance at the individual level beyond the SN/VTA. Accordingly, we excluded SN/VTA from non-midbrain DRT analyses, which also avoided any circularity. We also note that using within-study normalization of DRT as an alternative control for study effects left all inferences unchanged.

**MID Task.** In addition to the neurofeedback training, the participants of Study 2 (N = 25) performed a MID task that captures differences in adaptive and general

reward sensitivity. Participants performed this task after the neurofeedback training and after a break of approximately 45 min during which they left the scanner. In every trial of the MID task[35,80,88] first one of three cues appeared (Supplementary Fig. 7). One cue was associated with large reward (ranging from 0 to 2.00 CHF), one cue with small reward (0 to 0.40 CHF) and one cue with no reward. After a delay of 2.5 to 3 s, participants had to identify an outlier from three circles by pressing one of three buttons as quickly as possible. Depending on the cue, their response time and the correctness of the answer, participants gained an amount of money. Importantly, the use of large and small reward ranges enables investigation of individual differences not only in general reward sensitivity but also in how well the reward system adapts to different reward distributions, so-called adaptive reward coding[35]. To investigate whether reward sensitivity in the MID task results from (lack of) regulation success, we correlated degree of regulation transfer with total payment in the MID task. We found no significant relation between performance in the two tasks (Pearson's r = 0.0332, p = 0.88). Thus, we found little evidence to support the notion that reward sensitivity in the MID can be explained by preceding self-regulation success.

**MR Data pre-processing**. We despiked the functional data using the AFNI toolbox (National Institute of Mental Health; http://afni.nimh.nih.gov/afni). To account for differences in echo-planar-image (EPI) slice acquisition times we employed temporal interpolation of the MR signal, shifting the signal of the misaligned slices to the first slice[89] using FSL 5 (FMRIB Software Library, Analysis Group, FMRIB, Oxford, http://fsl.fmrib.ox.ac.uk). Furthermore, data were bias-field corrected using ANTs (Advanced Normalization Tools; http://stnava.github.io/ANTs), realigned using FSL 5, normalized to standard Montreal Imaging Institute (MNI) space using ANTs in combination with a custom scanner-specific EPI-template resulting in a 1.5 mm³ isotropic resolution and finally smoothed with a 6 mm full-width-half-maximum Gaussian kernel using FSL 5.

The spatial specificity control analyses (Supplementary Fig. 6 and Supplementary Table 7) suggest that our findings are not due to common physiological noise. To more directly account for noise, we additionally acquired physiological data in a subsample of participants. In the available subsample, neither changes in heart rate variability nor respiration were significantly correlated with VTA/SN activation during reward imagination (see details in refs. [14], Supplemental Material Supplementary Table 1, Supplementary Fig. 1). Here, we also used an image-based correction to account for physiological artefacts in all participants. Since physiological artefacts are most prominently present in cerebrospinal fluid and white matter due to the absence of BOLD effects, pulsations of the ventricles, and proximity to the large brain arteries (e.g., circle of Willis), we decided to use an established preprocessing procedure based on a principal component analysis (PCA) approach[90,91]. Specifically, we calculated the global mean and the first six components of a temporal principal component analysis on the cerebrospinal fluid and white matter signal. These six components were used as noise regressors in the first-level statistics (see section "MR Data analysis") in addition to the six motion parameters. Along with the pre-processing of the fMRI data, the SN/VTA mask used as ROI for the analysis was resliced into the dimensions of the functional data using SPM 12 (v6906, Wellcome Trust Centre for Neuroimaging, UCL, London, UK; http://www.fil.ion.ucl.ac.uk/spm/software/spm12/) within Matlab R2016b (Mathworks, Sherborn, MA, USA).

**MR Data analysis**

*Non-midbrain DRT correlation with midbrain DRT in standard and inverted feedback group (aim 1)*. The first question of this study asked whether the individual degree of successful SN/VTA neurofeedback transfer is associated with individual differences in the cognitive control network. To answer this question, we conducted a general linear model (GLM) on the single subject level including one block-wise regressor for the IMAGINE_REWARD condition and one for the REST condition with 190 timesteps (each condition comprised 9 onsets and lasted 20 s) for each of the four runs separately. Additionally, we modelled the first 5 TRs of every run as nuisance regressor and added also motion and physiological artefact regressors (see methods section for MR Data pre-processing) in the design matrix. In total the GLM consisted of fifteen regressors. We formed the contrast IMAGINE_REWARD-REST and compared it between Transfer and Baseline runs outside SN/VTA, i.e. non-midbrain DRT=(IMAGINE_REWARD-REST)$_{Transfer}$ − (IMAGINE_REWARD-REST)$_{Baseline}$.

At the group level, we tested for correlation of non-midbrain DRT with midbrain DRT. We ran these analyses in all voxels for both the standard and inverted feedback groups. To test for common and separate activity between the groups, we performed conjunction and disjunction analyses over the two group maps. Additionally, we performed a two-sample t-test to search for significant differences between groups. To identify activity within the cognitive control network, we used a cognitive control template based on the coordinates from a meta-analysis[36]. We created this template with fslmaths and spheres of 15 mm around all coordinates from the meta-analysis. In Supplementary Table 1 we identify regions of the cognitive control network where non-midbrain DRT correlates with midbrain DRT within the template. For statistical maps, we used an FWE-corrected cluster level threshold, p < 0.05 (cluster extent of 230 voxels) following a cluster-inducing voxel level threshold of p < 0.001 (uncorrected). In

addition, to test the functional specificity of our results, we performed a meta-analytic functional decoding analysis using the Neurosynth database (www.neurosynth.org). This relates the neural signatures of the cognitive control decoding network to other task-related neural patterns (Supplementary Fig. 5).

*Temporal difference coding during NF training (aim 2)*. The second question of the study asked whether successful neurofeedback performance was associated with a reduction in temporal difference coding during the training runs as captured by a classic reinforcement learning framework. There, learning corresponds to reducing prediction errors by adjusting predictions until they match experienced reward as much as possible. Mathematically, temporal differences can be viewed here as the first derivative of the observed BOLD signal in the midbrain. Traditionally, reinforcement learning theories compute the error term with the form defined by Sutton & Barto[50]:

$$\delta_t = R_{t+1} + \gamma V(S_{t+1}) - V(S_t) \quad (2)$$

$R_{t+1}$ is the next reward, $V(S_t)$ and $V(S_{t+1})$ are current and next reward predictions, $\gamma$ is a discount parameter (typically estimated to be 1 or close to 1 as t is short). In a task with continuous feedback (smiley height) like ours, $R_{t+1}$ and $V(S_{t+1})$ collapse, such that the error term becomes $\delta_t = R_{t+1} - V(S_t)$, which corresponds to the difference of subsequent feedback states (note that actions are not observable in neurofeedback training). In other words, the error term is the (continuously evolving) incongruence between the current feedback state, which incorporates participants' expectations about the upcoming feedback, and the next state, i.e. the actually presented feedback signal. As the highest available temporal resolution to compute the error term in our paradigm is one TR, we decided to approximate it with this resolution. However, we do not mean to imply that the brain is limited to that resolution. These temporal difference errors should be high when using unpracticed mental strategies at the beginning of the experiment. Over the time course of the neurofeedback training block, the temporal difference signal within the midbrain should decrease for participants who learn to successfully self-regulate the activity of their SN/VTA. For these participants, the change in height of the smiley should become more predictable.

To investigate temporal difference signals in non-midbrain regions in our paradigm, we constructed an additional GLM for the neurofeedback training runs and modelled the regulation conditions (IMAGINE_REWARD and REST) during neurofeedback training runs as event-related regressors for every TR (in contrast to the block-wise previous analysis). We parametrically modulated these regressors with a time-resolved continuous temporal difference term. This term was defined as difference between the current and the previous TR within the SN/VTA mask, i.e. the parametric modulator corresponded to the difference in the BOLD signal of the SN/VTA from IMAGINE_REWARD$_t$ –IMAGINE_REWARD$_{t-1}$. We formed this parametric modulator separately for the upregulate and the rest condition and analysed it run-wise to investigate brain regions (excluding SN/VTA) showing a correlation with temporal difference in SN/VTA. We then investigated (Supplementary Fig. 3) on the single subject-level if the relation to this temporal difference information decreased over time by using the difference between the parametric temporal difference modulator of the second neurofeedback training run and the parametric temporal difference modulator of the first neurofeedback training run, i.e. temporal difference error coding in later training minus earlier training. This difference should become negative as temporal differences decrease with learning and outcomes become more predicted. Finally, on the group level, we correlated this contrast (between-run difference in temporal difference coding) with non-midbrain DRT in a one-sample t-test to test for whole-brain associations between a decrease in temporal difference coding and successful midbrain self-regulation.

The results of this analysis, showing an association with decreasing temporal difference coding in the dorsolateral prefrontal cortex (dlPFC), inspired a functional connectivity analysis. Specifically, we investigated the functional impact of temporal difference coding in dlPFC on the SN/VTA using a psychophysiological interaction analysis using the gPPI v13 Toolbox[92] based on the MNI coordinate of dlPFC (x = 40, y = 10, z = 38) with a 5 mm sphere as seed region. We added activity from this seed region as physiological regressor to the original GLM and interacted it with both the IMAGINE_REWARD and REST regressors to form interaction regressors. Functional connectivity was calculated by contrasting the interaction terms IMAGINE_REWARD-REST between second and first neurofeedback training run. We then correlated this contrast with midbrain DRT. The results were focused to the SN/VTA region as target. For statistical maps, we used a whole-brain threshold of p < 0.001 (20 voxel extent).

*Add-on analysis Dynamic Causal Modelling (DCM)*. In addition to functional connectivity, we also investigated task-dependent effective connectivity between SN/VTA and dlPFC during the second neurofeedback training run related to the upregulation of the dopaminergic midbrain. The full DCM analysis is reported in Supplementary Note 3. The results suggest that successful self-regulation appears to benefit from some inhibitory modulation of SN/VTA by prefrontal cortex.

*Control analysis*. As an alternative form of learning, we tested for non-associative mechanisms and performed two additional analyses testing whether (de-)sensitization can explain how the repetition of mental strategies relates to midbrain DRT. In

particular, we assessed a linear parametric modulator for each timestep in the training sessions, both increasing or decreasing. The analysis of linear increase (sensitization) itself revealed a positive correlation with midbrain DRT within the left hippocampus ($x = -35$, $y = -26$, $z = -7$, uncorrected $p < 0.001$, cluster size = 30). In contrast, a linear decrease (desensitization) revealed a negative correlation with left ACC ($x = 43$, $y = 0$, $z = 14$, uncorrected $p < 0.001$, cluster size = 21). However, because the SN/VTA is more intimately related to associative than non-associative learning, we do not consider this analysis further.

*Relation between non-midbrain DRT and reward sensitivity in the MID Task (aim 3).* To address the third aim of the study, we investigated the relationship between reward processing in the MID task and the capacity to successfully regulate the SN/VTA in the transfer session of the neurofeedback experiment at the whole-brain level. In particular, we considered two contrasts in the MID task, (1) general reward sensitivity, defined as the sum of parametric modulators: small plus large reward; (2) adaptive reward coding, defined as the difference between parametric modulators: small minus large reward. Again, we used correlation analysis at the group level to determine whether these two contrasts are related with individual SN/VTA transfer success (non-midbrain DRT) in the neurofeedback task. Moreover, to assess the commonalities of the neural activities in the two different tasks, we performed a conjunction analysis of contrasts (1), (2) and the correlation of transfer-activity with non-midbrain DRT. For statistical maps, we used a whole-brain threshold of $p < 0.001$ (20 voxel extent).

**Additional behavioral measurements**. We analysed available information on external behavioral scores that might explain differences in the individual transfer success.

*Strategies.* All participants were introduced to five example strategies (see section about neurofeedback paradigm) they could use to upregulate brain activity but were also free to use their own strategies. At the end of the experiment, participants filled in a custom-made questionnaire on the strategies they used. To compare strategies between the groups, we used a $\chi 2$-test that assessed differences in the distribution of strategy usage. We did not observe any significant group differences in strategy use ($p = 0.9$), and therefore did not consider this measurement in any further analysis (Supplementary Table 8).

*Personality measures.* To control for the possibility that individual differences in behavior and personality were associated with individual differences in DRT, Study 2 measured: (1) Smoking status in number of cigarettes per day; (2) verbal IQ as determined by the Multiple Word Test (MWT[93]); (3) Positive and Negative Affect Score (PANAS) in the German version[94]; (4) attentional and nonplanning subscores of the Barratt Impulsivity Scale in the German version[95]. None of these variables correlated significantly with midbrain DRT in a Pearson's correlation test (all $p > 0.5$) and the correlations reported in the results section were robust to including the variables as covariates of no interest.

**Statistics and reproducibility**. For all of the reported analyses, we used the toolbox SPM 12 (v6906) within Matlab R2016b. All figures were created using bspmview v.20161108[96] and ggplot2 within R 3.4.1. All group-level analyses included an additional covariate for the dataset to account for potential global signal differences between studies. To estimate the robustness of our results, we performed bootstrapping analyses for all correlation analyses and summarized these in Supplementary Table 10. To validate the findings of this secondary data analyses, we performed a confirmatory analysis separating the two self-regulation conditions IMAGINE_REWARD and REST over the course of the study. Because the participants were instructed to perform mental calculations during REST, which is an active task, the results of these analyses indicate that the results of the main analysis involving the contrast (IMAGINE_REWARD-REST) are not simply driven by a decrease during REST (Supplementary Note 4).

**Reporting summary**. Further information on research design is available in the Nature Research Reporting Summary linked to this article.

## Data availability
Second level statistical map data supporting the findings of this study are available at https://identifiers.org/neurovault.collection:12684. All other source data, such as data extractions from regions of interest and mask files, for this paper and the Supplemental Material are provided with this paper in the file supplementary_data_1.zip. A reporting summary for this Article is available as a Supplementary Information file.

## Code availability
Matlab and R Code supporting this publication is publicly available at: https://github.com/lydiatgit/NFLearning_SNVTA_PublicRepo (https://zenodo.org/badge/latestdoi/391012761)[97].

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

## Acknowledgements

The authors would like to thank Silvia Maier and Stephan Nebe for comments on previous versions of the manuscript and fruitful discussions. This project was supported by the European Union's Horizon 2020 research and innovation program under the Grant Agreement No 794395 (to LH) and grant 100014_165884 and 10001C_188878 from the Swiss National Science Foundation (to PNT). MK received grant support from the National Bank Fellowship (McGill) and Swiss National Science Foundation (P2SKP3_178175).

## Author contributions

All authors contributed to critical review of the manuscript for important intellectual content, and final approval of submission of the manuscript for publication. J.S. and M.K. provided the data. L.H. and R.S. performed the analysis. L.H., P.N.T. and M.K. interpreted the results and wrote the paper. J.S., F.S., R.S. and M.H. revised the manuscript.

## Competing interests

The authors declare no competing interests.

## Additional information

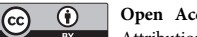

