## [Peer Review File · Communications Biology]

Reviewers' comments:

Reviewer #2 (Remarks to the Author):

I endorse publication.

Reviewer #3 (Remarks to the Author):

Broadly speaking, I think that the potential of neurofeedback training for clinical application would be greatly heightened by an increased understanding of the factors contributing to its success (i.e., transfer), and from that perspective, this study is likely to be of great interest to the field. I have been asked to assess the author's rebuttal to Reviewer #3 from the initial submission, and as such, I am not providing an entirely new review nor have I reviewed the original submission. I think the authors have effectively addressed most of the concerns, but I have a few remaining comments, only 1 of which is major.

Major comment

R3.2 (and more specifically 3.2.1): I share the same concern noted by Reviewer 3 and other reviewers regarding the prediction error framing. Broadly speaking, I believe the authors' clarifications on their operationalization of TD learning to be a huge improvement in that I have a clear understanding of the temporal difference calculation. I also very much appreciate the substitution of "temporal difference" for "prediction error" since it is a more accurate description of the calculation taking place. As I understand it, the error term computed in this operationalization amounts to $R_{t+1} - V(S_t)$, i.e., the difference in feedback between sequential TRs, something like a first-order derivative. And the high-level conceptual idea here is that as (more successful) participants learn to regulate their midbrain, they should have a better sense of what is working, thus forming more accurate predictions about upcoming feedback, thus shrinking the temporal difference (the operationalization of prediction error).

The authors write "As a consequence, they would be more likely to repeat the strategy, expect higher feedback next time and gradually learn how to keep the feedback signal high. Accordingly, in regulators the size of the temporal difference between feedback states should gradually decrease as the expected feedback increasingly converges with the actual feedback." And temporal difference here is the difference between sequential feedback. Thus, a hypothetical representation of that would be as follows (see hypothetical plots):

The contradiction arises here because in general, the association between midbrain activation and PE is that midbrain dopamine *signals* TD, and the authors themselves acknowledge this in the rebuttal in the passage they quote from Sutton & Barto, that "(E)xperimental evidence suggests that one neurotransmitter, specifically the neurotransmitter dopamine, signals RPEs, and further, that the phasic activity of dopamine-producing neurons in fact conveys TD errors." In the case of this study, it seems completely contradictory that midbrain activation remains high as a result of participants' increased success with self-regulation *and* signals TD (i.e., midbrain activity cannot simultaneously look like the plot on the left and the plot on the right in the hypothetical plots). Reviewer 3 — and I believe other reviewers have picked up on this contradiction as well — points out this contradiction in **R3.2.1**: "They define PEs as VTA activation TR to TR. I assume they don't observe decreasing PEs in midbrain, because the midbrain NF is "escaping" the adaptation as a result of (successful) regulation— (Or I assume not, because the slope during training predicts DRT. But, they should say this explicitly if they do actually observe this, to help explain the analysis choices.) So, they must find other brain regions that correlate with this signal." Thus, while on the surface, the connection to operant associative learning via prediction error is alluring, I think this framing adds

more confusion and contradiction rather than clarity.

I don't actually feel that the prediction framing is necessary for appreciation of the significance of the related findings. The finding that the decrease in the temporal difference with feedback training predicts transfer success is intuitive and a compelling finding in and of itself, without connecting it to prediction error. In fact, one way to look at it is that it may reflect, as the authors themselves say, the ability of successful participants to keep the signal high (like the hypothetical plot on the left). Reviewer 5's comment that this might be more reflective of stabilized learning rather than prediction error captures this idea. Furthermore, the relationship between dlPFC and temporal difference coding (and the subsequent functional connectivity finding) is also compelling evidence for dlPFC monitoring these changes, and builds nicely off of the cognitive control network findings of Aim 1. This is all to say that as a reader, I think backing off of the prediction error framing would strengthen rather than weaken this manuscript. If the authors insist on some type of mention of prediction error, I think the appropriate place for that would be to draw the connection in the discussion, but to clarify that the claim is not that midbrain is simultaneously being successfully up-regulated *and* coding the decreasing TD simultaneously, but rather that something like prediction error is possibly being tracked by the dlPFC.

Minor comment

R3.3: R3.3 stated "R3.3 Was the MID done before or after NF? I understood before, but this matters because in one case it predicts DRT, and in the other it may be a result of success, so should be stated. This relation is one of the paper's strengths and I find it more interesting than the stretch to reinforcement learning/PE." The authors' clarification on the order of tasks is noted and appreciated.

Nonetheless, I agree with Reviewer 3 that the order of these tasks matters to the interpretation of the correlation. This manuscript frames MID measures of reward processing as revealing something like *trait* reward sensitivity, and as such, the interpretation is that individual *trait* differences in reward sensitivity *predict* individual differences in midbrain self-regulation ability. However, Reviewer 3 points out that because the MID is done after the NF, the reward sensitivity in the MID could reflect some reward-sensitive *state* as a result of the preceding self-regulation success. I recognize that a 45min break in which the participants left the scanner makes this less likely than if the MID were done immediately after the NF, but I do think this alternative is worth explicit mention. I also believe that mentioning this alternative does not undermine the finding in any way, since the relationship between successful midbrain regulation and an independent measure of reward sensitivity is incredibly interesting in its own right.

Minor remark

R3.5 I think that clarification in the introduction and discussion on possible factors that contributed to the different findings regarding neurofeedback training efficacy (i.e., why transfer effects were observed in MacInnes et al, but not Sulzer et al or Kirschner et al) is useful. The choice of control group is an analytical and statistical factor, in that it may explain why the comparisons are significant. The use of VTA alone as a target for feedback (vs SN/VTA) is a methodological factor. The "best practice" references mentioned here focus on control group selection as well as feedback quality (denoising). One thing that might be interesting is whether there were any differences in the instruction of the participants. I don't know whether the authors would deem it useful to add to this manuscript, but just thought I would mention it in case it is of interest going forward. I could not find the instructions for the Sulzer et al or Kirschner et al studies, but I did find the MacInnes instructions: <https://www.cell.com/cms/10.1016/j.neuron.2016.02.002/attachment/ddbc338c-f251-404b-b648->

41e7e6bd64a1/mmc1.pdf and one thing I noticed was the instruction *"We ask that you stick with one strategy for the entire trial and if you find that strategy to be ineffective, feel free to try something else on the next trial."*

Reviewer #5 (Remarks to the Author):

The authors have done a tremendous job addressing previous reviewer comments. I particularly like that they changed the term prediction error to temporal distance. I now recommend publication

Response to Reviewer comments:

We warmly thank Reviewers #2 and #5 for their recommendation to publish our manuscript as is. Below, we address the remaining concerns of Reviewer #3. Please find our detailed responses (italic font) to the comments (bold). The changes in the manuscript are highlighted in red.

Reviewer #2 (Remarks to the Author):

I endorse publication.

We thank the reviewer for this endorsement.

Reviewer #3 (Remarks to the Author):

Broadly speaking, I think that the potential of neurofeedback training for clinical application would be greatly heightened by an increased understanding of the factors contributing to its success (i.e., transfer), and from that perspective, this study is likely to be of great interest to the field. I have been asked to assess the author's rebuttal to Reviewer #3 from the initial submission, and as such, I am not providing an entirely new review nor have I reviewed the original submission. I think the authors have effectively addressed most of the concerns, but I have a few remaining comments, only 1 of which is major.

Major comment

R3.2 (and more specifically 3.2.1): I share the same concern noted by Reviewer 3 and other reviewers regarding the prediction error framing. Broadly speaking, I believe the authors' clarifications on their operationalization of TD learning to be a huge improvement in that I have a clear understanding of the temporal difference calculation. I also very much appreciate the substitution of "temporal difference" for "prediction error" since it is a more accurate description of the calculation taking place. As I understand it, the error term computed in this operationalization amounts to $R_{t+1} - V(S_t)$, i.e., the difference in feedback between sequential TRs, something like a first-order derivative. And the high-level conceptual idea here is that as (more successful) participants learn to regulate their midbrain, they should have a better sense of what is working, thus forming more accurate predictions about upcoming feedback, thus shrinking the temporal difference (the operationalization of prediction error).

The authors write "As a consequence, they would be more likely to repeat the strategy, expect higher feedback next time and gradually learn how to keep the feedback signal high. Accordingly, in regulators the size of the temporal difference between feedback states should gradually decrease as the expected feedback increasingly converges with the actual feedback." And temporal difference here is the difference between sequential feedback. Thus, a hypothetical representation of that would be as follows (see hypothetical plots):

The contradiction arises here because in general, the association between midbrain activation and PE is that midbrain dopamine signals TD, and the

authors themselves acknowledge this in the rebuttal in the passage they quote from Sutton & Barto, that "(E)xperimental evidence suggests that one neurotransmitter, specifically the neurotransmitter dopamine, signals RPEs, and further, that the phasic activity of dopamine-producing neurons in fact conveys TD errors." In the case of this study, it seems completely contradictory that midbrain activation remains high as a result of participants' increased success with self-regulation and signals TD (i.e., midbrain activity cannot simultaneously look like the plot on the left and the plot on the right in the hypothetical plots). Reviewer 3 – and I believe other reviewers have picked up on this contradiction as well – points out this contradiction in R3.2.1: "They define PEs as VTA activation TR to TR. I assume they don't observe decreasing PEs in midbrain, because the midbrain NF is "escaping" the adaptation as a result of (successful) regulation– (Or I assume not, because the slope during training predicts DRT. But, they should say this explicitly if they do actually observe this, to help explain the analysis choices.) So, they must find other brain regions that correlate with this signal." Thus, while on the surface, the connection to operant associative learning via prediction error is alluring, I think this framing adds more confusion and contradiction rather than clarity.

I don't actually feel that the prediction framing is necessary for appreciation of the significance of the related findings. The finding that the decrease in the temporal difference with feedback training predicts transfer success is intuitive and a compelling finding in and of itself, without connecting it to prediction error. In fact, one way to look at it is that it may reflect, as the authors themselves say, the ability of successful participants to keep the signal high (like the hypothetical plot on the left). Reviewer 5's comment that this might be more reflective of stabilized learning rather than prediction error captures this idea. Furthermore, the relationship between dlPFC and temporal difference coding (and the subsequent functional connectivity finding) is also compelling evidence for dlPFC monitoring these changes, and builds nicely off of the cognitive control network findings of Aim 1. This is all to say that as a reader, I think backing off of the prediction error framing would strengthen rather than weaken this manuscript. If the authors insist on some type of mention of prediction error, I think the appropriate place for that would be to draw the connection in the discussion, but to clarify that the claim is not that midbrain is simultaneously being successfully up-regulated and coding the decreasing TD simultaneously, but rather that something like prediction error is possibly being tracked by the dlPFC.

We thank the reviewer for going into the mathematical theory to clarify the notion of prediction error. We would like to briefly note that both plots provided by the reviewer can be understood as expressions of prediction error according to the temporal difference rule and both responses occur in the same dopamine neurons in the same situation but at different time points. In this view, the left plot corresponds to the prediction error at the time of a reward predicting stimulus conditional on learning being successful. Accordingly, successful regulators should gradually expect reward from the imagery and strategies they use to upregulate the signal during training. However, we apologize if the reviewer understood that we identify regions processing such a *phasic* signal *during training* in Aim 1 – instead, for Aim 1 we identify regions outside midbrain *during transfer* where activity correlated with changes of *sustained* SN/VTA activity, similar to the signal MacInnes and colleagues (2016) focused on for the SN/VTA itself. Conversely, the right plot corresponds to the time of the reward, again conditional on learning being successful. Successful regulators should in this view experience gradually smaller temporal differences when their strategies yield the desired results. We identify regions processing such a signal in *non-midbrain regions* for Aim

2. Both signals can be understood as complementary expressions of one and the same mechanism: temporal difference computation, which at each moment in time tracks future reward (Figure 1). Accordingly, to the extent that a brain region implements temporal difference learning, one would expect both signals to be present in that brain region (which is indeed the case for single dopamine neurons). However, our paper focuses on different questions. Specifically, for both Aim 1 and Aim 2, we focus on regions outside the SN/VTA and leave aside the question whether both signals occur in SN/VTA. Moreover, our two analyses focus on different forms of responses (more sustained for Aim 1, more phasic for Aim 2) and phases of the task (transfer for Aim 1, training for Aim 2). We now clarify our approach further in the manuscript. Moreover, following the reviewer's advice, we also have fully adopted the temporal difference (rather than prediction error) framing whenever we describe our own work.

Figure 1. Cue- and outcome-related temporal difference signals develop in parallel and are captured by the same mechanism (reprinted from Schultz et al., 1997). Note that during association of cue with reward, response at outcome gradually decreases and response at cue gradually increases, in accordance with standard temporal difference learning rule.

We also note that dopamine neurons are heterogeneous, which allows for distinct forms of responses^{1,2}. Following on from this heterogeneity, different signals and functions might be reflected in the BOLD response. Specifically, while Aim 2 searched for non-midbrain associations with phasic SN/VTA signal changes, Aim 1 searched for non-midbrain associations with sustained SN/VTA signal changes. MacInnes and colleagues (2016) discuss the capability specifically of sustained self-modulation of VTA as follows:

“... we show that healthy individuals can learn to sustain VTA activation over a period of 20 s. This timescale is novel in the human literature and may relate to a controversial finding in the animal literature (Beeler et al., 2010; Fiorillo et al., 2003; Nishino et al., 1987; Niv et al., 2005; Schultz, 2007). Although a dichotomized view of dopamine neuron physiology as either tonic or phasic has guided the field for many years, converging evidence across methodologies (Fiorillo et al., 2003; Howett et al., 2013; Totah et al., 2013) suggests a third type of dopamine response profile: a sustained, ramping signal evident during anticipation of reward. This ramping signal is distinct from rapid phasic responses, but also differs from tonic signals theorized to reflect a summation of prior phasic events (Niv et al., 2007) or spontaneous firing (Goto et al., 2007); instead, it appears more consistent with sustained excitatory inputs. The VTA activations observed here are inconsistent with transient neural responses to external events and may be consistent with the sustained dopamine profile

described in the animal work. While our VTA BOLD signal did not demonstrate a ramping profile, the sustained nature of the signal converges with the neuronal signal observed in the animal work, in that both are novel profiles not accounted for by transient signals. Alternatively, it is also possible our results reflect summed sequential phasic responses as participants refresh strategies; BOLD imaging is currently unable to resolve this question. More fundamentally, however, these sustained signals were observed in the absence of external reward cues and therefore must have been elicited through internal representations.”

Although our data are based on a summary signal from the SN/VTA and the standard group comparison did not yield a significant transfer effect over the full group, our findings go in the same direction on individual transfer success levels as the findings of MacInnes and colleagues: Successful regulators acquire the capability to increase dopaminergic signals in a sustained fashion. With the second aim of our study, we go beyond this finding and investigate whether a temporal difference signal contributes to the learning of dopaminergic self-modulation. It has been shown that temporal difference signals can be captured in the BOLD response of regions within and outside of the midbrain^{3,4}. Temporal difference signals could appear either in different neurons within our target regions SN and VTA or in addition to sustained increase in the signal (Fiorillo et al., 2003). In any case, we search for associations outside the midbrain. This aspect has not been investigated so far and is novel in our analysis.

1. Fiorillo, C. D., Tobler, P. N. & Schultz, W. Discrete Coding of Reward Dopamine Neurons. *Science (1979)* **299**, 1898–1902 (2003).
2. de Jong, J. W., Fraser, K. M. & Lammel, S. Mesoaccumbal Dopamine Heterogeneity: What Do Dopamine Firing and Release Have to Do with It? *Annu Rev Neurosci* **45**, (2022).
3. Chase, H. W., Kumar, P., Eickhoff, S. B. & Dombrovski, A. Y. Reinforcement learning models and their neural correlates: An activation likelihood estimation meta-analysis. (2015) doi:10.3758/s13415-015-0338-7.
4. O’Doherty, J. P., Dayan, P., Friston, K., Critchley, H. & Dolan, R. J. Temporal Difference Models and Reward-Related Learning in the Human Brain. *Neuron* **38**, 329–337 (2003).

We have now thoroughly revised our terminology throughout manuscript and completely removed any prediction error framing whenever we talk about our work.

(p. 22, l. 433-440) “Therefore, we operationalized these temporal differences by subtracting the immediately preceding SN/VTA activity (prediction) from the present SN/VTA activity (outcome: Fig. S7). Specifically, we tested for a negative correlation of non-midbrain DRT with the difference in SN/VTA temporal difference signals between late and early training. In other words, for successful regulators, we expected to find a negative correlation between the decrease in midbrain temporal difference signal over the course of the neurofeedback training and activity in other brain regions. We found such a relation with gradually decreasing SN/VTA temporal difference signals in dlPFC (Fig. 5 and Table S5). “

(p. 27, l. 501-503) “Nonetheless, our finding yield the same direction on individual transfer success levels as the finding from MacInnes and colleagues (2016): successful regulators learn to increase dopaminergic signaling. “

(p. 29, l. 546-555) “Apart from the first aim of investigating a sustained form of dopaminergic responses during transfer, our second aim operationalized the first derivate of

the sustained modulation in dopaminergic midbrain as temporal difference signal. This allowed us to investigate a more phasic form of responses outside the dopaminergic midbrain during training. We found a reduction in the relation between SN/VTA decreasing temporal difference signal and DLPFC activity over the course of the neurofeedback training for successful regulators only, while this relation remained high for non-regulators. This finding suggests that a temporal difference-like signal might be tracked by the dIPFC, which is compatible with the notion that temporal difference error-driven reinforcement learning was more pronounced in regulators than non-regulators and provides empirical evidence for previous theoretical proposals on the principles of neurofeedback learning independent of feedback modality²⁵. “

Minor comment

R3.3: R3.3 stated “R3.3 Was the MID done before or after NF? I understood before, but this matters because in one case it predicts DRT, and in the other it may be a result of success, so should be stated. This relation is one of the paper’s strengths and I find it more interesting than the stretch to reinforcement learning/PE.” The authors’ clarification on the order of tasks is noted and appreciated.

Nonetheless, I agree with Reviewer 3 that the order of these tasks matters to the interpretation of the correlation. This manuscript frames MID measures of reward processing as revealing something like trait reward sensitivity, and as such, the interpretation is that individual trait differences in reward sensitivity predict individual differences in midbrain self-regulation ability. However, Reviewer 3 points out that because the MID is done after the NF, the reward sensitivity in the MID could reflect some reward-sensitive state as a result of the preceding self-regulation success. I recognize that a 45min break in which the participants left the scanner makes this less likely than if the MID were done immediately after the NF, but I do think this alternative is worth explicit mention. I also believe that mentioning this alternative does not undermine the finding in any way, since the relationship between successful midbrain regulation and an independent measure of reward sensitivity is incredibly interesting in its own right.

We thank the reviewer for following up on this alternative explanation. To shed brighter light on the issue, we performed an additional control analysis correlating the total payment participants received in the MID task with the degree of regulation transfer (Figure 2). This analysis revealed no significant relation (Pearson’s $\rho = 0.0332$, $p = 0.88$). Thus, we found little evidence to suggest that reward sensitivity in the MID results from preceding self-regulation success or lack thereof. We now describe this analysis in the manuscript.

Figure 2

(p. 12, l. 223-227) “To investigate whether reward sensitivity in the MID task results from (lack of) regulation success, we correlated degree of regulation transfer with total payment in the MID task. We found no significant relation between performance in the two tasks (Pearson’s $\rho = 0.0332$, $p = 0.88$). Thus, we found little evidence to support the notion that reward sensitivity in the MID can be explained by preceding self-regulation success. “

Minor remark

R3.5 I think that clarification in the introduction and discussion on possible factors that contributed to the different findings regarding neurofeedback training efficacy (i.e., why transfer effects were observed in MacInness et al, but not Sulzer et al or Kirschner et al) is useful. The choice of control group is an analytical and statistical factor, in that it may explain why the comparisons are significant. The use of VTA alone as a target for feedback (vs SN/VTA) is a methodological factor. The “best practice” references mentioned here focus on control group selection as well as feedback quality (denoising). One thing that might be interesting is whether there were any differences in the instruction of the participants. I don’t know whether the authors would deem it useful to add to this manuscript, but just thought I would mention it in case it is of interest going forward. I could not find the instructions for the Sulzer et al or Kirschner et al studies, but I did find the MacInnes instructions:

<https://www.cell.com/cms/10.1016/j.neuron.2016.02.002/attachment/ddbc338c-f251-404b-b648-41e7e6bd64a1/mmc1.pdf> and one thing I noticed was the instruction “We ask that you stick with one strategy for the entire trial and if you find that strategy to be ineffective, feel free to try something else on the next trial.”

Since the original instructions were given in German, an attachment would be helpful only for partial readership. We have now added a paragraph in the Supplement with a translated summary and an explicit comparison between the instructions. The three

studies all explicitly instructed participants to change their strategies between the regulation trials and figure out successful ones. Also, all explicitly explain in the instructions the delay in the feedback signal of several seconds and ask questions about the perceived self-performance afterwards. The main difference between the instructions consists in the proposed strategies – while MacInnes focusses on self-motivation, the other two studies refer to positive self-memories. Other differences arise from 1) MacInnes and colleagues instructing participants not only about the display of their current performance but also of their average performance and targeted performance and 2) Sulzer and colleagues and Kirschner and colleagues explicitly instructing participants to rest during “Rest” and only imagine positive memories during “Happy Time”.

(Supplemental Material, paragraph 2.16)

“2.16 Participants’ instructions in different studies

2.16.1 MacInnes and colleagues (2016)

Available from:

<https://www.cell.com/cms/10.1016/j.neuron.2016.02.002/attachment/ddbc338c-f251-404b-b648-41e7e6bd64a1/mmc1.pdf>

“During the ACTIVATE conditions you will see two upward-facing arrows, as well as a thermometer in the center of the screen. However, on these trials, the thermometer is reporting on-going activity within brain areas involved in reward and motivation. Your task is to try to get – and keep – the thermometer as high as possible. Try to do the same things you did during the Test task. That is, try to encourage yourself to increase your brain signal as much as possible. When the thermometer level is above zero it will be red, and when it’s below zero it will be blue. Since you are receiving feedback indicating how well you are doing, you can directly try to move the height of the thermometer with your brain. Try to encourage yourself to move that bar! You can adapt your strategies to discover which methods produce the greatest amount of activity. We ask that you stick with one strategy for the entire trial and if you find that strategy to be ineffective, feel free to try something else on the next trial.”

“During the ACTIVATE condition try your best to produce a mental state of heightened motivation. For example, try to encourage yourself that you can increase your own brain signal. It may help to think of this task as a fun game. Telling yourself positive phrases, such as “you can do it!” or “increase that signal!” or imagining personally relevant scenarios may be useful strategies for you. These are the sorts of mental states we’d like you to try to produce. Importantly, you don’t have to limit yourself to these phrases. Feel free to try anything that you think will be motivating. What motivates you may be different than what motivates me, so you should think about things that will best motivate you personally.”

2.16.2 Sulzer and colleagues (2013)

Available from <http://www.ncbi.nlm.nih.gov/pubmed/23791838>

“The participants were instructed to attempt to gain self-control over the region of the brain activated by novel rewarding stimuli. Examples of rewards, such as food, romantic or sexual imagery, time with family and friends, and achievements were suggested. After our pilot studies had shown that romantic or sexual imagery was most effective in volitionally

controlling the BOLD signal in the SN/VTA, the participants were informed of these results but were allowed to adapt their strategy according to neurofeedback success. The participants were asked to maximize the height of a vertically moving ball on the screen, representing their brain activity, when cued, and informed that the scanner acquisition time as well as the hemodynamic effects would cause approximately 5 s of delay between thought and the feedback signal.”

2.16.3 Kirschner and colleagues (2018)

Available from [https://www.thelancet.com/journals/ebiom/article/PIIS2352-3964\(18\)30472-9/fulltext](https://www.thelancet.com/journals/ebiom/article/PIIS2352-3964(18)30472-9/fulltext)

“Outside the scanner, participants were instructed about the goal of the experiment, i.e. to gain self-control over the reward-related brain regions by imagining non-drug related rewarding stimuli. To assess the ability of generating vivid mental imagery, we used an adapted version of the Prospective Imagery Task (PIT) [36,37]: we provided a list of five potentially rewarding sceneries/topics (i.e., positive experiences with family and friends, professional achievements, romantic or sexual memories, hobbies, delicious food including positive scents) plus two individually defined topics, which they rated according to speed (how rapidly mental images can be generated), vividness, and detail on a scale from 1 to 10. Only the three best ranked topics were used during scanning.”

2.16.4 Comparison

The instructions used by Sulzer and colleagues and by Kirschner and colleagues were in German. In addition to the summary from the papers above, participants were explicitly instructed to change their thoughts between the regulation trials and figure out successful ones. This is similar to the instructions used by MacInnes and colleagues. Also, all instructions explicitly explained the delay in the feedback signal of several seconds and all studies asked questions about the perceived self-performance afterwards.

The main difference between the instructions used by the three studies concerns the suggested choice of mental strategies – while MacInnes and colleagues focus on self-motivation for the neurofeedback task, the other two studies refer to positive self-memories. Other differences arise from 1) MacInnes and colleagues instructing participants not only about the display of their current performance but also of their average performance and targeted performance and 2) Sulzer and colleagues and Kirschner and colleagues explicitly instructing participants to rest during “Rest” and only imagine positive memories during “Happy Time”.

(p. 27, l. 490-492) “The instructions given to the participants differed mainly in the recommended strategies (see detailed comparison in Supplement 2.16).”

Reviewer #5 (Remarks to the Author):

The authors have done a tremendous job addressing previous reviewer comments. I particularly like that they changed the term prediction error to temporal distance. I now recommend publication

We thank the reviewer for the recommendation.

REVIEWERS' COMMENTS:

Reviewer #3 (Remarks to the Author):

The rebuttal, clarifications, and revised manuscript are convincing and excellent.